

# Factors influencing the distribution of woody plants in tropical karst hills, south China

Gang Hu[1,2,*], Zhonghua Zhang[2,*], Hongping Wu[1] and Lei Li[1]

[1] Ministry of Education Key Laboratory for Ecology of Tropical Islands, College of Life Sciences, Hainan Normal University, Haikou, Hainan, China
[2] Key Laboratory of Wildlife Evolution and Conservation in Mountain Ecosystem of Guangxi, School of Environmental and Life Sciences, Nanning Normal University, Nanning, Guangxi, China
* These authors contributed equally to this work.

Corresponding author
Lei Li, lei-li@126.com

## ABSTRACT

The seasonal rainforests distributed across the tropical karst hills of south China are of high biodiversity conservation value and serve many important ecosystem functions. However, knowledge surrounding distribution patterns of woody plants in tropical karst hills remains limited. In this study, we surveyed the distribution of families, genera and species of woody flora at four slope positions (depression, lower slope, middle slope, and upper slope), and analyzed the influence of topographic and soil variables on the distribution of woody plants in the tropical karst hills of south China. Forty forest plots (each 20 m × 20 m) contained 306 species of woody plants with a diameter at breast height (DBH) ≥1 cm, representing 187 genera and 66 families. As slope increased, the number of families increased slowly, and the number of genera and species followed a concave-shaped trend, with the lowest number of genera and species in the lower slope position. Differences in species composition were significantly stronger between slope positions than within slope positions. The topographic and soil variables explained 22.4% and 19.6%, respectively, of the distribution of woody plants, with slope position, slope degree, soil potassium and soil water content as the most significant variables. The results of generalized linear mixed model analysis showed that total $R^2$ of fixed effects on variation of woody species richness was 0.498, and rock outcrop rate and soil total phosphorus were the best fitting effects. Our results help to explain the community assembly mechanism and to inform management and protection strategies for species-rich seasonal rainforests in the karst area.

## INTRODUCTION

The distribution of tree species in forests and the influencing mechanisms have always been a central topic in ecology (*Du et al., 2017*). The species composition and distribution of plant communities are results of the interaction between plant species and their environment, and how environmental variables influence plant communities is often

scale-dependent (*Siefert et al., 2012*). Generally speaking, climate and soil parent material are closely related to distribution patterns of plant communities at a regional scale (*e.g.*, province and country), while the topographic and soil variables play critical roles in determining this pattern at a local scale (*e.g.*, hill, county, and district) (*Fang & Lechowicz, 2006*; *Jarema et al., 2009*; *Song et al., 2010*). In the mountainous ecosystem, variables such as topography and soil nutrient availability lead to high habitat heterogeneity at different altitudes, resulting in a varied distribution pattern of plant communities along the environmental gradient (*Zhang et al., 2013*; *Birhanu et al., 2021*). Currently many of the world's forests are experiencing habitat degradation and biodiversity loss due to increasing pressure from human disturbance and climate change (*Guo et al., 2017*; *Forzieri et al., 2022*). Studies on the relationship between forest communities and the environment not only contribute to a broader understanding of tree distribution and community assembly, but they also guide restoration and reconstruction efforts in degraded forest ecosystems (*Rahman et al., 2021*).

Karst is a distinctive topography formed when rainfall and groundwater act on carbonate bedrock, such as limestone or dolomite (*Jiang, Lian & Qin, 2014*). Globally, karst area occupies about 15% (approximately 22 million km$^2$) of all land area (*Chen et al., 2020*). China has the largest karst area of any country in the world, with a total area of 3.443 million km$^2$. The karst area in southwest China is one of the largest contiguous karst areas in the world. The provinces of Guangxi, Guizhou and Yunnan have the highest concentrations of karst landforms (*Guo et al., 2013*), with the karst area of Guangxi covering 41.57% (98,700 km$^2$) of this province (*Li et al., 2022*). The karst ecosystem is unique for its sparsely and unevenly distributed soils (usually alkaline), weak water retention capacity and low threshold for resisting external disturbance. In terms of appearance, the karst area has a varied terrain where peaks gather into clusters shaped like cones or towers, and where the round or oval depressions that form inside the peak-clusters are closed. This landform creates highly heterogeneous terrains and an abundance of microhabitats that allow different plant species to selectively utilize different spaces. This leads to the highly diverse communities and abundance of endemic species found in karst hills (*Waltham, 2008*; *Zhang et al., 2013*; *Du et al., 2017*).

The seasonal rainforests of south China, known for their highly diverse flora, are one of the representative forest types in the tropical karst area. Nonggang National Nature Reserve (NNNR) in Guangxi province is a large-area karst seasonal rainforest with a complex community structure and an abundance of endemic species. It is home to one of the 14 key biodiversity hotspot areas in China (*Huang et al., 2013*; *Wang et al., 2014*). Earlier studies have completed a primary investigation of taxonomic diversity among the plant communities in NNNR (*Liang & Mo, 1982*; *Liang et al., 1985*). Other studies have analyzed the structure and dynamics of the tree population (*Lv et al., 2004*) and the restoration dynamics of the karst vegetation (*Deng et al., 2004*). Since a forest monitoring plot (500 m × 300 m) was established in NNNR in 2011, the studies on karst seasonal rainforests have turned to focus on spatial distribution patterns of plant communities

(*Wang et al., 2014*), species diversity patterns (*Guo et al., 2018*), litter dynamics (*Guo et al., 2020*), and tree mortality patterns (*Guo et al., 2021*) at a 15-ha plot scale. These studies have enhanced our understanding of the structure and dynamics of karst forests in the NNNR.

Although species associations with topography and soil factors at local scales have been reported in tropical forests (*Webb & Peart, 2000*; *John et al., 2007*), the current understanding of how environmental variables affect the plant distribution in heterogeneous karst hills is still limited (*Geekiyanage et al., 2019*; *Liu et al., 2021*). Vegetation is likely to be governed by various environmental variables, and any variations in species composition are often linked to various environmental gradients (*Toure & Ge, 2014*). As a key topographic variable, slope position can affect microclimate, soil properties, species composition, and even ecosystem function (*Zeng et al., 2014*; *Dearborn, Danby & Kikvidze, 2017*). In the karst hills, soil depth and water availability strongly decrease from the depression towards the upper slope (*Zhang et al., 2020*). Compared with the non-karst region, the unique two-dimensional hydrological structure of the karst area can lead to serious loss of surface soil and water (*Geekiyanage et al., 2019*), which may cause the vegetation to experience geological drought and may lead to great differences in the availability of resources at different slope positions on the hill. Previous studies have shown that slope position can also affect the physicochemical characteristics of soil in karst hills (*Liu et al., 2017*; *Liang et al., 2018*; *Yang et al., 2019*). Moreover, the hydraulic characteristics and differences in the stomatal regulation of water content among different plant species in karst hills are closely related to changes in slope position (*Zhang et al., 2020*). Therefore, the distinctive hydrological process formed by the peak-cluster depressions creates high heterogeneity in environmental variables, such as soil water and nutrition, at different slope positions. This heterogeneity may underlie the spatial distribution of plant communities in karst hills at a local scale.

Perturbation of any one of the variables within the highly complex and interactive karst system is likely to upend the others (*Toure & Ge, 2014*). This makes it obvious to tell that soil and topographic properties are interrelated in their association with plant species, but defining this association and to what degree it exists is a challenge. Therefore, uncovering the relationships of these effects with the plant species will allow for researchers to better predict species' responses to changes in environmental variables in the karst ecosystem. In this study, we observed woody flora and determined environmental variables at four slope positions (depression, lower slope, middle slope and upper slope) in the karst hills in NNNR. The objectives of this study were (1) to investigate the distribution pattern of woody flora in the tropical karst seasonal rainforest at different slope positions; and (2) to identify the effect of environmental variables (topography and soil) and geographic distance on the distribution of woody plants in the tropical karst hills in NNNR. We hypothesized that complex plant community distributions in the heterogeneous karst habitat are directly influenced by topographic and soil variables, among others. The information learned in this study will guide plant conservation and ecological restoration in future efforts in tropical karst areas.

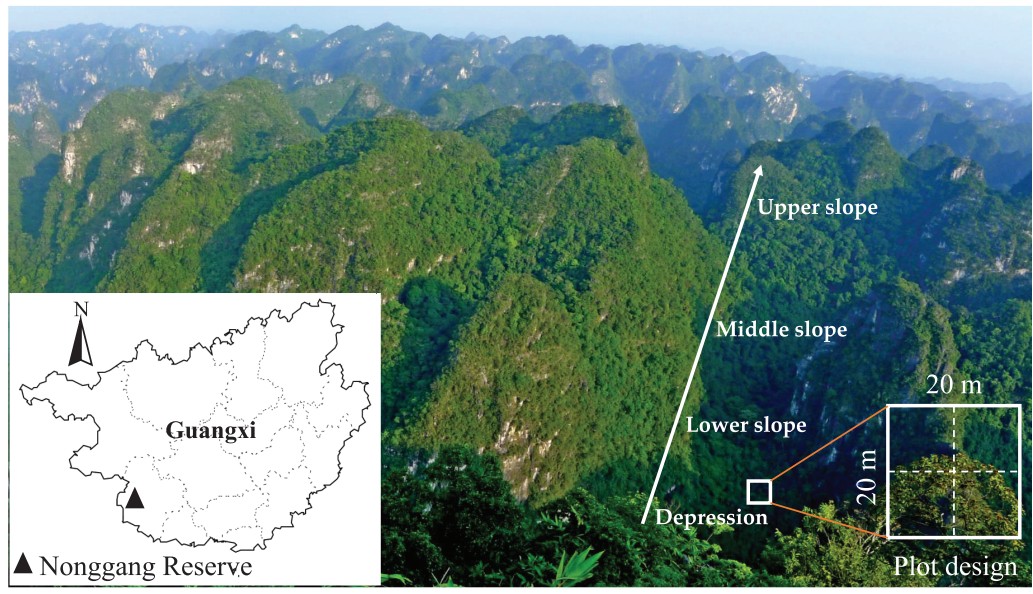

**Figure 1 Location of Nonggang National Nature Reserve and karst peak-cluster depression landscape.**

## MATERIALS AND METHODS

### Study area

The study site was located in the NNNR (22°13′56″–22°39′09″N, 106°42′28″–107°04′54″E) in the southwest of the Guangxi province of south China, which also borders Vietnam (Fig. 1). The total area of the reserve was 10,077 ha. The landform is typified by typical karst peak-cluster depression landscapes with elevation ranging from 118.2–680.1 m, and the average relative height difference between the depression and the peak was about 350 m. The study area had a tropical monsoon climate with annual precipitation ranging from 1,200–1,500 mm and a maximum precipitation of 2,043 mm. The annual average temperature was 22 °C, and the annual accumulated temperature was 7,400–7,800 °C. The soil was characterized as brown calcareous soil with abundant Ca, low water storage capacity and relatively high pH (Bai et al., 2019). There were a total of 1,752 species of vascular plants, from 810 genera and 184 families, in NNNR. This composition included 39 Chinese endemic fern plant species, 101 Guangxi endemic seed plant species, 278 species of karst endemic plants, 33 species of rare and endangered plants and 73 species of key protected wild plants of Guangxi (Huang et al., 2013). The representative vegetation was tropical karst seasonal rainforest, which included rare and endangered plants such as *Excentrodendron tonkinense*, *Deutzianthus tonkinensis*, *Caryota obtusa*, *Garcinia paucinervis*, and *Cephalomappa sinensis*. This tropical karst seasonal rainforest was home to 28 species of nationally-protected animals including *Trachypithecus francoisi*, *Trachypithecus poliocephalus*, and *Moschus berezovskii*.

## Plot survey

The karst hills are often extremely steep with extensive carbonate rock outcrops. Thus, it is very difficult to investigate forest communities in NNNR. We selected four slope positions (depression, lower slope, middle slope, and upper slope) in which to establish forest plots (each 20 m × 20 m) based on the changes in hill height on four hills spaced at intervals of more than 800 m apart (Fig. 1). Ten plots were established at each slope position with intervals of at least 30 m (most >100 m) from one plot to the next on the same hill. The large intervals between each plot helped to avoid spatial autocorrelation effects (*Moradi, Attar & Oldeland, 2017*). A total of 40 plots were established in the study area. Each was divided into four 10 m × 10 m subplots for the survey. All woody plants with a diameter at breast height (DBH) ≥1 cm in were tagged, measured and identified to species. At the same time, environmental variables, including elevation, slope aspect, and rock outcrop rate, were recorded for each plot. The slope degree and rock outcrop rate were estimated by visual inspection, while the others, elevation, longitude and latitude, were measured in the center of the plot using GPS receivers (GPSMAP60CSx; Garmin China Shanghai Co., Ltd., Shanghai, China). For more details about the plots see Table S1.

## Soil sampling and analysis

Soil samples from 0–20 cm in depth were collected from each plot according to the five-spot-sampling method, where four samples are collected at the corners and one in the center of the plot, and then the five samples are uniformly mixed to form one sample (*Li et al., 2021*). These soil samples were then air-dried and filtered (2-mm sieve) in the laboratory before the analysis of chemical properties could begin. Soil pH was determined *via* potentiometry; soil organic matter (SOM) was measured according to a method of potassium dichromate oxidation-outer heating; soil total nitrogen (STN) was measured using the micro-Kjeldahl method; soil total phosphorus (STP) was measured *via* NaOH fusion and the Mo-Sb colorimetric method; soil total potassium (STK) was measured *via* NaOH fusion and flame photometry; soil exchangeable calcium (SECa) was extracted with DTPA and measured using atomic absorption spectrometry. Soil water content (SWC) was measured in the field using a soil multi-parameter monitoring system (WET-2; Delta-T Devices Ltd, Cambridge, United Kingdom).

## Data analysis

The family, genus and species name of the woody plants were checked using the R package "plantlist" (*Zhang, 2017*). To determine the dominant species of each plot across the four slope positions, relative and absolute frequency, density, and dominance of each species were calculated. From the relative metrics, the importance value index was calculated for each species (IVI), genus (GVI) and family (FIV). IVI, GVI and FIV were calculated as follows: IVI = (relative density of species + relative frequency of species + relative dominance of species)/3; GVI = (relative diversity of genus + relative density of genus +

relative dominance of genus)/3; and FIV = (relative diversity of family + relative density of family + relative dominance of family)/3 (*Jiménez-Paz et al., 2021*).

ANOSIM (analysis of similarities) was used to check for significant differences in species composition between different slope positions (*Clarke, 1993*; *Warton & Wright, 2012*). ANOSIM was calculated using a matrix of Bray–Curtis dissimilarity based on data of woody species abundance, and was performed in the "vegan" package in R 4.0.3 (*R Core Team, 2020*). A *P*-value with a significance level of *P* < 0.05 was obtained using a permutation test (999 randomizations) and was checked by FDR (false discovery rate). The calculation formula was as follows:

$$R = \frac{r_b - r_w}{0.25[n(n-1)]}$$

where $r_b$ represents mean rank of dissimilarities between different slope positions; $r_w$ represents mean rank of dissimilarities within slope positions; $n$ represents the number of plots; $R$ represents community similarity with a range from −1 to 1, a greater $R$ indicates a stronger difference between slope positions, $R > 0$ means that the difference was stronger between slope positions than within slope positions, and *vice versa*.

The spatial autocorrelation of environmental variables in each plot was checked by using the "ape" package, and any significant autocorrelation (Moran's I = 0.057–0.332, $P < 0.05$) was not founded. In our study, detrended canonical correspondence analysis (DCCA) was used to evaluate the effects of the environmental variables on the distribution of woody plants. DCCA method can avoid the potential arch effect of canonical correspondence analysis (CCA) and can illustrate the relationship between vegetation characteristics and environment variables more obviously than an ordination diagram alone (*Qiu & Zhang, 2000*; *He et al., 2019*). The vegetation data matrix consisted of species abundance in each plot. The environmental data matrix consisted of 11 environmental variables, including four topography variables (slope position, slope degree, slope aspect, and rock outcrop rate) and eleven soil variables (soil pH, SOM, STN, STP, STK, SECa, and SWC). Species with a weight value greater than 30% (*i.e.*, species with relative abundance over 30% in each plot) were selected for plotting on the DCCA ordination diagram. A Monte Carlo permutation test (999 randomizations) was performed to determine the significance of the eigenvalues. For slope position, type variables 1, 2, 3 and 4 were used to indicate depression, lower slope, middle slope and upper slope, respectively (*Qiu & Zhang, 2000*; *Yu et al., 2013*). For the slope aspect, an azimuth angle within 0°–360° was transformed into a TRASP (transformation of aspect) index. The method of calculating the TRASP index is attributed to *Yu et al. (2013)*. DCCA ordination was carried out using Canoco 5.0 software (*ter Braak & Šmilauer, 2012*).

Variance partitioning analysis (VPA) was performed on the environmental variables (topography and soil) and geographic distance to determine the proportion of different types of variables explaining the variation in plant communities. The geographic distance was determined using latitudinal and longitudinal coordinates of each plot, and principal coordinates of neighbor matrices (PCNM) were used as the geographic distance

explanatory variable (*Borcard & Legendre, 2002*). The above analysis was carried out using the "vegan" package.

There were random factors among plant communities at different slope positions, so the generalized linear mixed model (GLMM) was used to analyze the driving factors of species richness. Firstly, the Spearman correlation analysis was tested for exploring collinearity between variables, and the results showed that the correlation coefficient between variables was small (Table S6), so the selected variables were retained for subsequent analysis. Secondly, the residual normality and homogeneity of variables was tested, and the dispersion of the dependent variable (*i.e.*, species richness) was tested so the appropriate regression model could be selected. Finally, the stepwise regression method was used to select the best fitting model based on the AIC values. In GLMM, the fixed effects (*i.e.*, independent variables) were the environmental variables. The relationship between the independent and dependent variables differed based on the type of plot, which may be one of the reasons why the residuals were not homogeneous. Thus, the type of plot was used as a random factor and fitted to a random intercept in the model. The dependent variable was overdispersed, so a negative binomial regression was chosen in GLMM (*Ver Hoef & Boveng, 2007*). The GLMM was performed in the "lmerTest" package. In addition, the total $R^2$ of the GLMM was calculated using the "MuMIn" package, and the interpretation rate (*i.e.*, $R^2$) of each independent variable was calculated using the "glmm.hp" package (*Lai et al., 2022*). The association between fixed effects and species richness was presented with forest plots, which were created using the "ggforestplot" package.

The differences in the number of families, genera and species for woody species and the environmental variables between different slope positions were analyzed statistically by one-way analysis of variance (ANOVA). If the data showed homogeneity of variance, Tukey's HSD test was used for multiple comparisons; if the variance was unequal, Games-Howell test was used for analysis. The above analysis was performed in SPSS 26.0 (SPSS Inc., Chicago, IL, USA).

## RESULTS

### Distribution of woody flora at different slope positions

Three hundred and six species of woody plants with DBH ≥ 1 cm from 187 genera and 66 families were recorded in the plots. In the depression there were 122 species from 98 genera and 41 families. In the lower slope there were 105 species from 83 genera and 43 families. In the middle slope there were 122 species from 87 genera and 43 families. In the upper slope there were 149 species from 112 genera and 50 families (Fig. 2). The number of families, genera and species varied significantly with the gradient of slope position (Fig. 2). As slope position increased, the number of families increased slowly, and the number of genera and species followed a concave-shaped trend, with the lowest number of genera and species in the lower slope (Fig. 2). Euphorbiaceae, Moraceae, Fabaceae, Rubiaceae and Phyllanthaceae were the most species-rich families, and *Ficus*, *Diospyros*, *Mallotus*, *Litsea* and *Tarenna* were the most species-rich genera in the plots (Tables S2–S4). The dominant families at all four slope positions were Euphorbiaceae, Phyllanthaceae, Malvaceae and Fabaceae. The slope positions did not share the same dominant genera and species (Tables

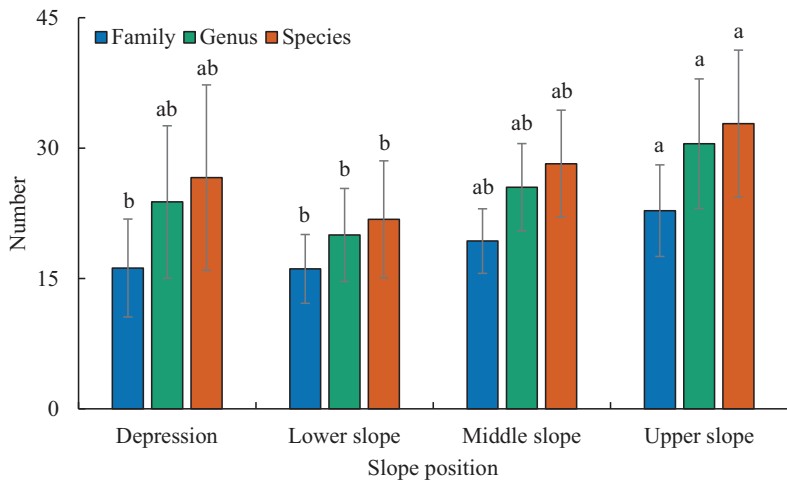

**Figure 2 The number of families, genera, and species at different slope positions.** Different lowercase letters indicate significant difference between different slope positions ($P < 0.05$). Error bar indicates Mean ± Standard deviation (SD).

**Table 1 The results of ANOSIM analysis for comparing differences in species composition between different slope positions.**

| Pairs | R | P |
|---|---|---|
| Depression *vs.* lower slope | 0.495 | 0.001 |
| Depression *vs.* middle slope | 0.881 | 0.001 |
| Depression *vs.* upper slope | 0.878 | 0.001 |
| Lower slope *vs.* middle slope | 0.402 | 0.001 |
| Lower slope *vs.* upper slope | 0.556 | 0.001 |
| Middle slope *vs.* upper slope | 0.445 | 0.001 |

S2–S4). The sum of the IVs of the top 10 families, genera and species showed a single-peaked curve along the gradient of increasing slope position, and the peak and minimum values appeared in the lower slope and upper slope, respectively (Tables S2–S4).

The ANOSIM results showed that the between group was higher than the other groups, meaning that the differences were stronger between slope positions than within slope positions, and they were significant too ($P = 0.001$) (Fig. S1). The woody species varied greatly between slope positions, with the largest difference between the middle slope and the depression (R = 0.881), then between the upper slope and the depression (R = 0.878), and the slightest difference was between the middle slope and the lower slope (R = 0.402). The differences between different slope positions were significant ($P = 0.001$) (Table 1).

**The effect of environment variables on distribution of woody plants**

The Monte Carlo test showed that all ordination axes were significant ($P = 0.001$), which meant that the correlations were also significant. The eigenvalues of the first and second axes were 0.717 and 0.354, respectively. The coefficients describing the species-environment correlations were 0.978 and 0.954. The first two axes in the DCCA

**Table 2 Correlation coefficients between environmental variables and the first two axes of DCCA ordination and ordination summary.**

| Environmental variables and summary of ordination | Axis 1 | Axis 2 |
|---|---|---|
| **Environmental variables** | | |
| Slope position | −0.931 | −0.032 |
| Slope degree | −0.802 | 0.142 |
| Rock outcrop rate | −0.652 | 0.477 |
| Slope aspect | −0.137 | −0.118 |
| Soil pH | −0.337 | 0.382 |
| Soil exchangeable calcium | −0.640 | 0.471 |
| Soil organic matter | −0.578 | 0.259 |
| Soil total nitrogen | −0.427 | 0.608 |
| Soil total phosphorus | 0.679 | 0.609 |
| Soil total potassium | 0.785 | 0.130 |
| Soil water content | 0.780 | −0.043 |
| **Summary of DCCA ordination** | | |
| Eigenvalues | 0.717 | 0.354 |
| Explained variation (cumulative) | 8.150 | 12.170 |
| Pseudo-canonical correlation | 0.987 | 0.954 |
| Explained fitted variation (cumulative) | 22.480 | 33.590 |

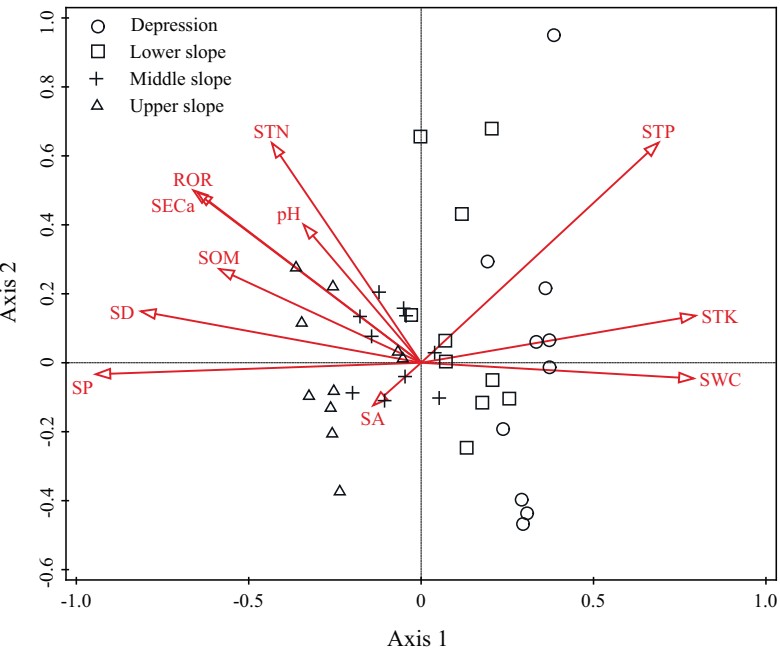

**Figure 3 Detrended canonical correspondence analysis (DCCA) ordination diagram of environmental variables and plots.** SA, slope aspect; SD, slope degree; SP, slope position; ROR, rock outcrop rate; pH, soil pH; SECa, soil exchangeable calcium; SOM, soil organic matter; STN, soil total nitrogen; STP, soil total phosphorus; STK, soil total potassium; SWC, soil water content.

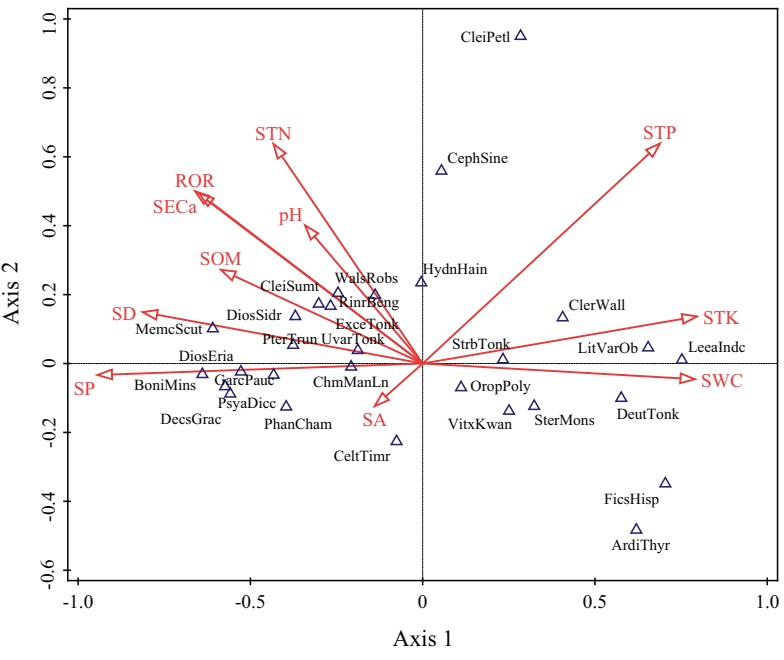

**Figure 4 Detrended canonical correspondence analysis (DCCA) ordination diagram of environmental variables and main woody species.** SA, slope aspect; SD, slope degree; SP, slope position; ROR, rock outcrop rate; pH, soil pH; SECa, soil exchangeable calcium; SOM, soil organic matter; STN, soil total nitrogen; STP, soil total phosphorus; STK, soil total potassium; SWC, soil water content. ArdiThyr, *Ardisia thyrsiflora*; BoniMins, *Boniodendron minus*; CeltTimr, *Celtis timorensis*; CephSine, *Cephalomappa sinensis*; ChmManLn, *Champereia manillana* var. *longistaminea*; CleiPetl, *Cleistanthus petelotii*; CleiSumt, *Cleistanthus sumatranus*; ClerWall, *Clerodendrum wallichii*; DecsGrac, *Decaspermum gracilentum*; DeutTonk, *Deutzianthus tonkinensis*; DiosEria, *Diospyros eriantha*; DiosSidr, *Diospyros siderophylla*; ExceTonk, *Excentrodendron tonkinense*; FicsHisp, *Ficus hispida*; GarcPauc, *Garcinia paucinervis*; HydnHain, *Hydnocarpus hainanensis*; LeeaIndc, *Leea indica*; LitVarOb, *Litsea variabilis* var. *oblonga*; MemcScut, *Memecylon scutellatum*; OropPoly, *Orophea polycarpa*; PhanCham, *Phanera championii*; PsyaDicc, *Psydrax dicocca*; PterTrun, *Pterospermum truncatolobatum*; RinrBeng, *Rinorea bengalensis*; SterMons, *Sterculia monosperma*; StrbTonk, *Streblus tonkinensis*; UvarTonk, *Uvaria tonkinensis*; VitxKwan, *Vitex kwangsiensis*; WalsRobs, *Walsura robusta*.

ordination explained 33.59% of the variation in the relationship between the distribution of woody plants and the environmental variables (Table 2).

The ordination diagram showed an obvious tendency of plots to transition from one slope position to the next. Along the first axis, from right to left, the plots changed from the depression to upper slope (Fig. 3). The first axis was highly positively correlated with SWC, STK and STP, but highly negatively correlated with SP, SD, ROR, SECa and SOM; the second axis showed a strong positive association with STP, STN and SECa (Table 2). As shown in Fig. 4, *Leea indica*, *Litsea variabilis* var. *oblonga*, *Deutzianthus tonkinensis*, *Ficus hispida*, *Ardisia thyrsiflora* and *Sterculia monosperma* were located on the far left of the ordination diagram, which indicates that these species adapted to habitats with high values of SWC, STK and STP. *Vitex kwangsiensis*, *Streblus tonkinensis*, *Orophea polycarpa*, *Hydnocarpus hainanensis*, *Cleistanthus sumatranus* and *Excentrodendron tonkinense* were located in the middle of the diagram, meaning that they adapted to habitats with relatively

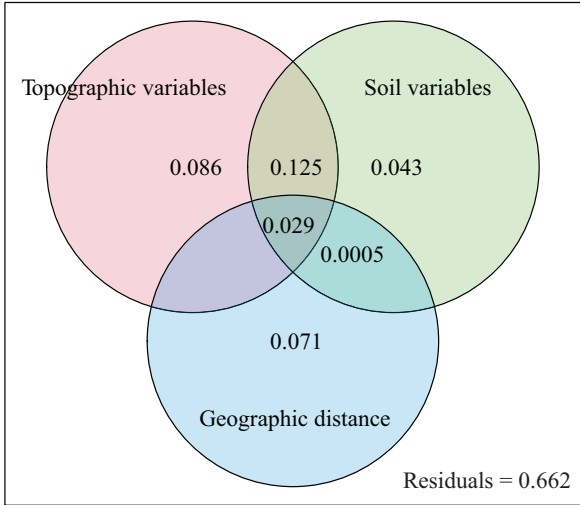

Values <0 not shown

**Figure 5 Venn diagram representing the variation partitioning between soil, topographic, and geographic variables.**

**Table 3 Generalized Linear Mixed models showing the effects of environmental variables on species richness.**

| Environmental variables | Standardized coefficient | R² | P-value |
|---|---|---|---|
| Slope aspect | 0.030 | 0.009 | 0.476 |
| Slope degree | −0.060 | 0.016 | 0.506 |
| Slope position | 0.020 | 0.068 | 0.823 |
| Rock outcrop rate | −0.060 | 0.078 | 0.477 |
| Soil pH | 0.007 | 0.002 | 0.889 |
| Soil exchangeable calcium | 0.030 | 0.009 | 0.652 |
| Soil organic matter | 0.009 | 0.004 | 0.868 |
| Soil total nitrogen | 0.020 | 0.031 | 0.759 |
| Soil total phosphorus | −0.200 | 0.185 | 0.019 |
| Soil total potassium | −0.080 | 0.083 | 0.244 |
| Soil water content | 0.080 | 0.013 | 0.236 |

moderate water conditions and levels of soil nutrients. Other species, such as *Decaspermum gracilentum*, *Memecylon scutellatum*, *Diospyros siderophylla*, *Diospyros eriantha*, *Psydrax dicocca* and *Boniodendron minus*, were located on the right side of the ordination diagram, indicating that these species adapted to habitats with high values of soil pH, STN, SECa, SOM, ROR and SD.

The explanatory power of different environment variables on species distribution was quantitatively analyzed. The topographic variables explained 22.4% ($P = 0.001$) of the species distribution, including 8.6% of variation caused by purely topographic variables (Fig. 5). The soil variables explained 19.6% ($P = 0.001$), including 4.3% of variation caused by purely soil variables. The geographic distance explained 8.4% of species composition,
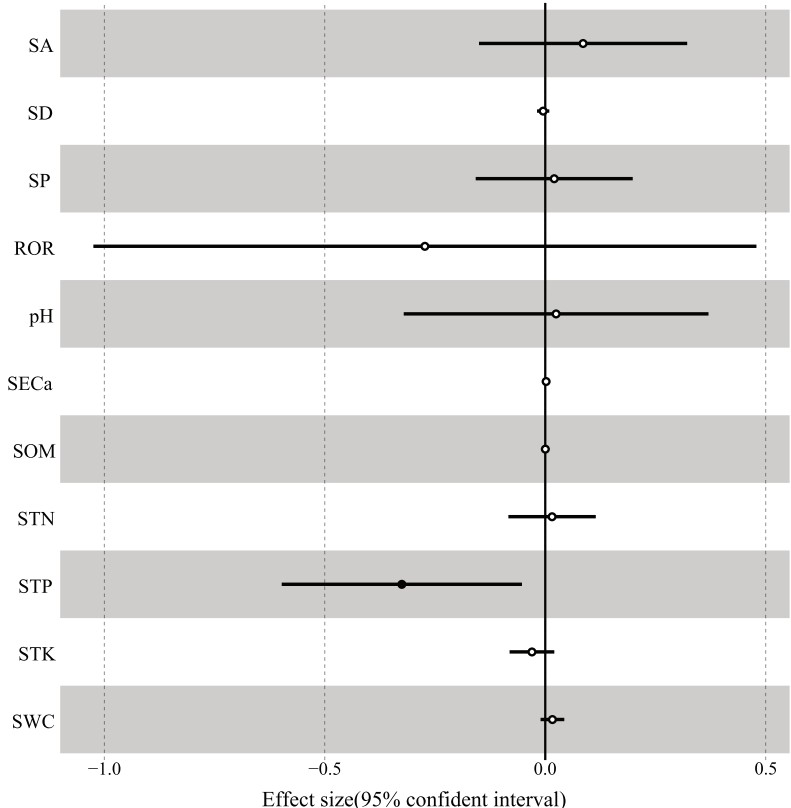

**Figure 6 The associations of environmental variables with species richness.** SA, slope aspect; SD, slope degree; SP, slope position; ROR, rock outcrop rate; pH, soil pH; SECa, soil exchangeable calcium; SOM, soil organic matter; STN, soil total nitrogen; STP, soil total phosphorus; STK, soil total potassium; SWC, soil water content.

**Table 4 Results of model selection for predicting species richness.**

| ID | Fixed effects | logLik | AICc | ΔAICc | Weight |
|----|---------------|--------|------|-------|--------|
| 1 | ROR + STP | −133.659 | 279.100 | 0.000 | 0.030 |
| 2 | SD + STP | −133.819 | 279.400 | 0.320 | 0.025 |
| 3 | SD + STK + STP | −132.597 | 279.700 | 0.660 | 0.021 |
| 4 | ROR + SECa + STP | −132.735 | 280.000 | 0.930 | 0.019 |
| 5 | ROR + STK + STP | −132.755 | 280.100 | 0.970 | 0.018 |
| 6 | ROR + SP + STP | −132.803 | 280.200 | 1.070 | 0.017 |
| 7 | ROR + SOM + STP | −132.806 | 280.200 | 1.070 | 0.017 |
| 8 | SD + SECa + STP | −133.014 | 280.600 | 1.490 | 0.014 |
| 9 | SD + SP + STP | −133.138 | 280.800 | 1.740 | 0.012 |
| 10 | pH + SD + STP | −133.232 | 281.000 | 1.920 | 0.011 |

Note:
Only the top 10 models with the best fitting effects are presented; AICc, Akaike information criterion for small samples; ΔAICc, Difference between the AICc of a given model and that of the best model; Weight, Akaike weights; ROR, rock outcrop rate; STP, soil total phosphorus; SD, slope degree; STK, soil total potassium; SP, slope position; SECa, soil exchangeable calcium; SOM, soil organic matter; pH, soil pH.

but did not have significant effect ($P = 0.284$). The environmental variables and geographic distance explained 33.8% of the variation, and topographic and soil variables jointly explained 22.5% of the distribution of woody plants. The unexplained portion accounted for 66.2% (Fig. 5).

According to the results of GLMM, the total $R^2$ of fixed effects on the variation in woody species richness was 0.498, and soil total phosphorus had the strongest interpretation rate ($R^2 = 0.185$) and was significantly negatively correlated with species richness (Table 3, Fig. 6). In addition, soil total potassium ($R^2 = 0.083$), rock outcrop rate ($R^2 = 0.078$), and slope position ($R^2 = 0.068$) had strong explanatory power for species richness (Table 3). The best-fitting model included rock outcrop rate and soil total phosphorus (Table 4), and the other variables of soil total potassium, soil exchangeable calcium, and slope position also were deemed to have strong fitting effects on species richness (Table 3).

## DISCUSSION

This study counted 306 species (from 66 families and 187 genera) of woody plants in 40 small-size plots (each 0.04 ha) established at four slope positions in the tropical karst hills in NNNR. Surprisingly, the species number in our study was more than that of the number of woody species from a survey in a 15-ha-size plot (500 m × 300 m) (223 species belong to 56 families and 157 genera) of the same kind of forest as is in NNNR (Wang et al., 2014). This indicates that forest inventories based on many small plots scattered over heterogenous habitats in tropical karst hills may encompass greater woody species diversity than would a survey in fewer, larger plots because the multiple small plots may cover a greater range of environmental gradients in karst habitats, including patches of rare habitat or areas containing locally rare species.

It is well known that the distribution of floristic composition in the mountains is affected by hydrothermal conditions along the elevation gradient (Zhang, Hu & Ni, 2013). Although the elevation of peak-cluster depressions in the karst hills was relatively low (most were under 600 m), there was strong topographical differentiation in soil and water availability along the hillslope (Zhang et al., 2011; Chuyong et al., 2011; Zhang et al., 2020). In this study, topographic and soil variables varied greatly at different slope positions (Table S5). Correlation analysis indicated that species richness was significantly negatively correlated with soil total potassium and total phosphorus, and significantly positively correlated with slope position (Table S6). Further analysis by GLMM showed that soil total phosphorus had the strongest explanatory power for species variation, while soil total phosphorus and rock outcrop rate were the best fitting effects for species richness (Tables 3 and 4). Thus, soil total potassium, total phosphorus, slope position and rock outcrop rate were main factors driving the variation in woody species richness. Karst hills are usually shaped like towers and cones (see Fig. 1), so areas in the upper slope are narrow, but they contain various microhabitats (rock cliff, stony gully and soil surface, etc.) that offer living spaces for a variety of plants differing in their habitat preferences (Zhang et al., 2013). Moreover, some species, such as *Viburnum triplinerve*, *Walsura robusta*, *Tirpitzia sinensis*, *Diospyros saxatilis*, and *Pistacia weinmanniifolia*, are exclusively distributed in the upper slope, where the habitats are arid, highly alkaline, and the soil layers are thin (*Geekiyanage*

*et al., 2019*; *Jiang et al., 2021*). This concentration of a variety of conditions augments species richness in this upper slope position (*Guo et al., 2017*). *Huang et al. (2016)* thought that almost no species can have an absolute advantage in a community based in the fragile habitats of the upper slope. This may help explain how a greater number of species are able to co-exist. Soil erosion due to strong precipitation in the rainy season leads to vertical heterogeneity in soil thickness, where soil moves downward along the slope over time (*Peng & Wang, 2012*). Therefore, the deep and moist soil available in karst depression areas may allow for the growth of more plants. The lower slope stands on the edge of the depression, so it usually also has good soil nutrient and water conditions. The woody flora *Excentrodendron tonkinense* and *Cephalomappa sinensis* absolutely dominate the lower slope (*Xiang et al., 2013*; *Wang et al., 2014*), so the area has a relatively low woody flora species richness. Compared with other slope positions, the middle slope is steeper and poorer in soil nutrient and water availability. A few tree species with strong adaptability, such as *Cleistanthus sumatranus*, dominate the middle slope (*Wang et al., 2014*). In this study, the content of soil total potassium, total phosphorus and water were significantly higher in the depression than in the middle and upper slope (Table S5). Phosphorus plays an important role in maintaining the growth and metabolism of plants (*Zeng et al., 2016*), so high soil total phosphorus content is conducive to the growth of plants. However, soil total phosphorus became a limiting factor of species richness in this study. The high soil $Ca^{2+}$ content, pH value and water content in karst areas may have enhanced soil phosphorus fixation, thereby forming as insoluble apatite that reduced the availability of phosphorus (*Tan et al., 2019*). Meanwhile, excessive soil potassium content can reduce the biomass of fine roots (*Wright et al., 2011*), so potassium is typically the main limiting nutrient for karst plants. Soil total potassium was concentrated at low slope positions due to the eluviation and plant accumulation effects, resulting in the damage of woody plants and the reduction of species richness. This is consistent with the conclusion that plant species richness is higher when soil nutrients like phosphorus and potassium are at medium or low levels (*Potts et al., 2002*). On the middle slope, the level of woody flora species richness is between that of the lower slope and the upper slope. The distinctive habitat conditions of different slope positions underscores the habitat specificity of plant species (*Chuyong et al., 2011*). Therefore, different slope positions in the topical karst hills of NNNR greatly differ in their composition of woody species.

Topography is commonly correlated with important environmental variables, especially the groundwater regime and the soil physical and chemical properties, which impact soil conditions that in turn have varied effects on vegetation distribution (*Toure & Ge, 2014*). Moreover, slope position impacts the microenvironment by changing climatic factors (*i.e.*, light and temperature), the distribution and availability of both soil nutrients and water content through gravity and hydraulic power, and the spatial pattern and diversity of vegetation in mountain areas (*Seibert, Stendahl & Sørensen, 2007*; *Du et al., 2015*). Our results suggest that topography and soil attributes are the most important factors affecting the distribution pattern of woody species. The depression in karst hills has high soil water content (Fig. 3) and a relatively shorter solarization time. Previous studies have shown that shade-tolerant and moisture-loving plants dominate in the depression (*Wang*

*et al., 2014*). Species such as *Ficus hispida*, *Leea indica* and *Deutzianthus tonkinensis* were the most common in the depression (Table S3). The upper slope has a desiccative environment with low soil water content and strong sunlight, so woody plants suffer strong stresses there. Thus, drought-tolerant and sun-loving plants, such as *Tirpitzia sinensis*, *Boniodendron minus* and *Diospyros siderophylla*, were common in the upper slope (Table S3). The woody species with different habitat preferences were distributed across slope positions according to the soil water content, which decreased as the hill elevation increased in our study (Table S5). *Zhang et al. (2020)* found that plant leaves in karst areas showed higher drought resistance as elevation increased. Our results also demonstrate a transition in species as slope position increases in tropical karst hills in the NNNR from shade-tolerant and moisture-loving to drought-tolerant and sun-loving. The result of DCCA showed that topographic and soil variables had significant effects on vegetation distribution, where different types of woody plant species were distributed at different slope positions. This supports our hypothesis that complex plant communities are shaped by environmental variables that influence the distribution of plant species.

Our results from VPA show that the coordination between the topographic and soil variables significantly positively affected the distribution of woody plants (Fig. 5). In this study, the soil variables of soil water content, total potassium and total phosphorus correlated negatively with topography variables, such as slope position, slope degree and rock outcrop rate, while the soil variables of soil pH, soil exchangeable calcium, total nitrogen and organic matter associated positively with the topographic variables (Figs. 3 and 4). Therefore, the topographic variables often influence vegetation distribution by affecting both soil nutrients and water content (*Zhang, Hu & Ni, 2013*). Slope position, slope degree, soil water content and total potassium correlated strongly with the first axis of DCCA. This indicates that they were the main driving factors for woody plant composition (Fig. 4). Further, soil total phosphorus, exchangeable calcium, total nitrogen and organic matter were also shown to be strong drivers of plant distribution (Table 2). *Du et al. (2015, 2017)* found that the elevation, slope degree, slope aspect, slope position, rock outcrop rate, soil organic matter, total phosphorus, available nitrogen and available phosphorus significantly influenced variation in species composition in an evergreen and deciduous broad-leaved mixed forest in a subtropical karst area of southwest China. Similarly, *Zhang, Hu & Ni (2013)* also reported that the major factors affecting vegetation distribution in subtropical karst forests of southwest China were elevation, slope degree, rock outcrop rate, soil total nitrogen, total phosphorus, total potassium. Soil has a direct effect on plant distribution, and topography often affects plant distribution by indirectly affecting soil conditions (*Du et al., 2015*). For example, the higher rock outcrop rate at high slope positions leads to a higher content of soil exchangeable calcium, which in turn promotes the growth of calcium-loving plants, such as *Excentrodendron tonkinense*, *Garcinia paucinervis*, and *Cinnamomum saxatile*. These species are karst endemic plants, and they were mainly concentrated in the high slope positions in the karst hills (*Guo et al., 2017*). Our results demonstrate that the topographic (*i.e.*, slope position, rock outcrop rate, slope degree and slope aspect) and the soil variables (*i.e.*, soil total potassium, total

phosphorus exchangeable calcium and total nitrogen) are the main factors influencing the vegetation distribution in tropical karst forests in NNNR.

The explanatory power of environmental variables on vegetation variation is determined by the complexity of vegetation. The more complex the vegetation, the lower the explanatory power of environmental variables (*Du et al., 2015*; *Shen & Zhang, 2000*). In this study, environmental variables explained 22.5% of the distribution of woody plants in a tropical karst area. *Shen & Zhang (2000)* found that environmental variables explained 22.25% of the spatial pattern of the evergreen broad-leaved forest at Dalaoling in the Three Gorges, while *Zhang et al. (2012)* found that environmental variables explained 70.7% of the variation of *Stipa breviflora* communities in the Inner Mongolian region. These studies suggest that the explanatory power of environmental variables on plant communities increases from forest to desert steppe habitats, as the complexity of vegetation is reduced. Our study demonstrates that the ability of environmental variables to explain the distribution of plant communities in a tropical seasonal rainforest of south China was relatively lower (22.5%) than in the results from *Zhang, Hu & Ni (2013)* (51.06%) and *Du et al. (2015)* (53.3%), where the study areas were located in an evergreen and deciduous broad-leaved mixed forest in a subtropical karst area of southwest China. *Ou et al. (2014)* also reported a relatively low explanatory power (46.38%) in a subtropical karst species-rich forest. The unexplained part of environmental variables in this study may be related to human disturbance, biological interactions or other factors, such as dispersal limitation (*Song, Liu & Liu, 2009*; *Zhang et al., 2013*; *Du et al., 2015*).

## CONCLUSIONS

Across 40 plots in the tropical karst hills, 306 species from 187 genera and 66 families of woody plants were found, with the lowest number of species in the lower slope and the largest number of species in the upper slope. The habitat of the upper slope was severe and plants situated there were subject to strong habitat stress, so conservation efforts in tropical karst forest should account for this stress when dealing with the entire plant community. The environmental variables of soil total potassium, total phosphorus, slope position and rock outcrop rate were the main factors driving the variation in woody species richness. The species composition varied greatly along the environmental gradient, with the differences in species composition stronger between slope positions than within slope positions. The environmental variables of slope position, slope degree, soil water content and total potassium significantly influenced the spatial distribution of woody plants. The topographic and soil variables had significantly effects on woody plant distribution and explained 22.4% and 19.6%, respectively, while the geographic distance explained 8.4% of the variation and did not have significant influence on woody plant distribution. The environmental variables varied along the hillside, promoting the niche differentiation of plants and causing significant differences in woody plant composition between slope positions. Therefore, we suggest that different vegetation restoration measures should be taken at different slope positions to best promote the growth of plants depending on the local topographic and soil nutrient conditions. Our findings are noteworthy for the

development of sustainable management policies for conservation and restoration in the topical karst area.

### Funding
This work was supported by the Guangxi Natural Science Foundation (2021GXNSFFA196005; 2021GXNSFAA196024), the National Natural Science Foundation of China (31960275; 31760128). The funders had no role in study design, data collection and analysis, decision to publish, or preparation of the manuscript.

### Grant Disclosures
The following grant information was disclosed by the authors:
Guangxi Natural Science Foundation: 2021GXNSFFA196005 and 2021GXNSFAA196024.
National Natural Science Foundation of China: 31960275 and 31760128.

### Competing Interests
The authors declare that they have no competing interests.

### Author Contributions
- Gang Hu conceived and designed the experiments, performed the experiments, analyzed the data, prepared figures and/or tables, authored or reviewed drafts of the article, and approved the final draft.
- Zhonghua Zhang conceived and designed the experiments, performed the experiments, analyzed the data, prepared figures and/or tables, authored or reviewed drafts of the article, and approved the final draft.
- Hongping Wu conceived and designed the experiments, analyzed the data, authored or reviewed drafts of the article, and approved the final draft.
- Lei Li conceived and designed the experiments, performed the experiments, authored or reviewed drafts of the article, and approved the final draft.

### Data Availability
The raw data are available in the Supplemental Files.

### Supplemental Information
Supplemental information for this article can be found online at http://dx.doi.org/10.7717/peerj.16331#supplemental-information.

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
