# Peer review of "Factors influencing the distribution of woody plants in tropical karst hills, south China"

_PeerJ, doi:10.7717/peerj.16331_

## Round 0.1 · original submission · Major Revisions

The manuscript is reviewed by two referees and both recommended major revision. Major revision is required according to each comments of the two reviewers, please.

Reviewer 1 ·

Basic reporting

The manuscript explores the relationship between soil characteristics and slope location with tree species distribution, for a very unique a important biome (karstic vegetation). The manuscript is very well written and easy to follow. I have some suggestions that might help make the manuscript more interesting and go beyond the descriptive scope the authors have use to present their work. First and foremost, I felt that the article needs a broader context, even if at least in a few short sentences, although shortly explored at the end of the discussion, the manuscript could benefit from inserting the study within a wider context. Plant species are expected to respond to soil and slope, why would it be different for karst vegetation?

Experimental design

Great desing , and samplig effort, I would suggest complementary statistical analysis, to help elucidate process involved in the distribution of trees found.

Validity of the findings

I think that there is a novelty of the study but is not very well explored by the authors. I feel the used a more descriptive strategy, than fully exploring the process underpinning the distribution of trees. I suggest a few more specified analysis,like testing which of the soil characteristic is more associated to the tree richness or abundance of species. please read the attached file.

Additional comments

Specific comments:
Lines 82 – 86: Is this exclusive of Karts areas or in general for all vegetation patter, what is known in a broader context about such variables?

Lines 189 – 191: This is the same as what fig.2 shows. Maybe present the overall mean values, as there does not seem to be statistical differences.

Lines 202 – 204: loose sentence, and not very intuitive. I would suggest being more straight forward. This sounds like the direct description of the ANOSIM test, I guessing something in the line of:
The slope position varied greatly, with middle slope and depression between the most different ones... Or something along that line. Which were the greatest difference?

Lines 213 – 214: I have difficulties obtaining this information from the Fig. 3. It does not come clear. What do you mean when stating: "obvious tendency of transition? What is transition between depression to upper slope?
Lines 217 – 220: Couldn't this be plotted instead? I would be more easy to infer, than having to look to two plot, to perceive this trends.
Lines 229 – 231: This tells me about the power or accuracy of the analysis. But not so much about how the environmental variables explained tree diversity. How did the soil characteristic vary between slopes? are there correlations? For instance, upper slopes seem to be characterized with lower water retention, hence species adapted to lower water content are those dominant in the upper slopes. This understanding allows you to infer about the biological process structuring the tree community (just an idea)

Lines 251: replace the word “with”, maybe “like” or “similar to”

Lines 260 – 263: A plot showing how these variables relate could be useful, since you have the data to do so.

Lines 274 – 277: Some of this relationship could be more clearly presented. Although I agree they are partly presented in the Fig.3 and Fig. 4. but not in a very straight forward way.

Lines 279: SWC, I suggest using the actual words, or at least, water content as is more straight forward, reader won’t have to go back to remember what the letters mean.

Line 284 – 285: Excellent, that can be tested and plotted with the data you have.

Line 292: I strongly suggest using the actual words, rather than the initials, as they make it difficult to follow the ideas, without having to go back and forth.

Lines 303 – 305: Several studies have shown that soil and slope affect tree diversity, this is fairly expected and known for many vegetation types across the world. It would be nice to see a broader discussion about, and specific implications. So far, you just describe that there are correlation, but infer little about the process.

Lines 332 – 335: This conclusion repeats what has been said throughout the manuscript. It would be nice to talk about the implications in terms of preserving lower vs. upper slopes. What about the depression, which are in more pressure in China? A bit of context in which your results are inserted would greatly improve the importance of the assessment made.

Annotated reviews are not available for download in order to protect the identity of reviewers who chose to remain anonymous.

·

Basic reporting

This is a very interesting and well-written manuscript on plant community correlates in a rare and possibly threatened ecosystem in China. I greet the authors for the huge amount of sampled species, product of spectacular field work. I believe this work is an important contribution to community ecology, and it has much potential to generate a basis for decision making in conservation. However, I made some notations mainly about the sampling design and data analysis methods that need to be improved so that the manuscript can be accepted for publication in the next steps. I apologize for any mistake in English writing, which is not my first language.

Experimental design

Additional information is required for full understanding of the sampling design. According to Table S1, most plots have a slope greater than 0º. Therefore, it is possible that there is altitudinal variation within plots. If this is the case, it is not clear how elevation values were measured (average?), nor how the authors dealt with variation in other environmental factors associated with elevation within each plot. Please clarify.

Validity of the findings

I have some questions about your analytical workflow, which potentially can affect the validity of the findings:

- You have 40 plots and 11 predictor variables. How can one be sure that your DCCA models are not overfitted because of excessive parameters? A plausible solution to deal with this possible bias is to apply some model selection method.

- Why have you categorized slope instead of using continuous values as a predictor variable?

- It is not clear what are the units of measurement of each predictor variable (maybe you could include this information in the raw data table), but apparently you have variables expressed in different units. In this case, how did you deal with the different scales in which the variances of each predictor differ among the plots? Perhaps it would be good to scale the predictor variables.

- It is not clear why you selected the 30% top species of relative abundance. Please provide a justification.

- You have not presented any control of spatial autocorrelation. How can one be sure that the effects of the predictor variables are not confused with the geographical distance between plots?

Additional comments

I have some specific comments, identified by line numbers:

41 - 43: The opening sentence seems confused. What is the difference between plant communities and vegetation? Isn't vegetation cover a relevant component of the environment? I suggest rewriting more clearly.

43 - 46: The areas covered by the regional and local scales you mentioned are not obvious. You may add an area size estimate to each of these scales in parentheses.

45: It is not if you are talking about patterns in general, or a particular pattern.

95 - 100: Further details are needed here. What exactly are your hypotheses and expectations? I suggest you create a more robust background for your hypothesis testing.

119: Replace the tropical... by The tropical...

126: For a didactic presentation on the sampling design, I suggest including the location and perhaps a schematic design of the plots in Fig.1.

136 - 137: Please, provide more details and a bibliographic reference for the five-spot-sampling method.

157: Which metric was used to calculate dissimilarities? Species abundance?

193: It does not seem that a concave-shaped trend is evidenced by Fig. 2.

202 - 204: According to the leveling of the boxes shown in Fig. S1, some comparisons do not seem to result in significant differences (e.g. depression vs. lower slope). Please check if the coefficients informed in Table 1 are correct.

234: You could include geographical distance on the diagram shown in Fig. 5.

---

## Round 0.2 · Minor Revisions

Minor revision is required to improve the manuscript.

·

Basic reporting

Dear editor and authors,

I recognize the authors' effort to improve the manuscript following the recommendations of the first round of review, and I am satisfied with most responses in the rebuttal letter. However, there is still a point to be discussed. No evidence was presented supporting that conclusions are not biased by spatial autocorrelation. The response provided by the authors about this topic is subjective, and does not solve the question. Additionally, the authors argued that they did not include geographic distance in the VPA analysis (diagram shown in Figure 5) because "the distances between slope positions in Karst Hills are relatively close". But this is also too subjective, and the role of geographic distance in this study system remains obscure. I insist that clear evidence on the effects of spatial autocorrelation are necessary for transparency and reliability. For example, an autocorrelation test (e.g. Moran´s I) could be applied on the residuals derived from the main models, and illustrated with a spatial correlogram with different classes of geographic distance, and distance could be added to the diagram shown in Figure 5.

All the best.

Experimental design

All comments were made in Basic Reporting.

Validity of the findings

All comments were made in Basic Reporting.

Additional comments

All comments were made in Basic Reporting.

---

## Round 0.3 · accepted · Accept

The manuscript is recommended to b accepted in PeerJ. The authors revised the manuscript according to comments, thanks